# Using a Smartwatch to Record Precordial Electrocardiograms: A Validation Study

**DOI:** 10.3390/s23052555

**Published:** 2023-02-25

**Authors:** Joske van der Zande, Marc Strik, Rémi Dubois, Sylvain Ploux, Saer Abu Alrub, Théo Caillol, Mathieu Nasarre, Dirk W. Donker, Eline Oppersma, Pierre Bordachar

**Affiliations:** 1IHU Liryc, Electrophysiology and Heart Modeling Institute, Fondation Bordeaux Université, F-33600 Pessac, France; 2Cardio-Thoracic Unit, Bordeaux University Hospital (CHU), F-33600 Pessac, France; 3Cardiovascular and Respiratory Physiology, TechMed Centre, University of Twente, 7522 NB Enschede, The Netherlands; 4Cardiology Department, Clermont Universite, Université d’Auvergne, Cardio Vascular Interventional Therapy and Imaging (CaVITI), Image Science for Interventional Techniques (ISIT), UMR6284, CHU Clermont-Ferrand, F-63003 Clermont-Ferrand, France

**Keywords:** ambulatory electrocardiography, Apple Watch, cardiac arrhythmias, mobile health, reliability, wearables

## Abstract

Smartwatches that support the recording of a single-lead electrocardiogram (ECG) are increasingly being used beyond the wrist, by placement on the ankle and on the chest. However, the reliability of frontal and precordial ECGs other than lead I is unknown. This clinical validation study assessed the reliability of an Apple Watch (AW) to obtain conventional frontal and precordial leads as compared to standard 12-lead ECGs in both subjects without known cardiac anomalies and patients with underlying heart disease. In 200 subjects (67% with ECG anomalies), a standard 12-lead ECG was performed, followed by AW recordings of the standard Einthoven leads (leads I, II, and III) and precordial leads V1, V3, and V6. Seven parameters (P, QRS, ST, and T-wave amplitudes, PR, QRS, and QT intervals) were compared through a Bland–Altman analysis, including the bias, absolute offset, and 95% limits of agreement. AW-ECGs recorded on the wrist but also beyond the wrist had similar durations and amplitudes compared to standard 12-lead ECGs. Significantly greater amplitudes were measured by the AW for R-waves in precordial leads V1, V3, and V6 (+0.094 mV, +0.149 mV, +0.129 mV, respectively, all *p* < 0.001), indicating a positive bias for the AW. AW can be used to record frontal, and precordial ECG leads, paving the way for broader clinical applications.

## 1. Introduction

Within just a few years, smartwatches have evolved from devices that show the time to comprehensive health trackers. The adoption of wearables in medicine has expanded worldwide with a rapidly growing number of consumers and with new features capable of real-time monitoring of health parameters [1,2]. Selected smartwatches, including Apple Watch (AW), Withings, and Samsung smartwatches, can record and transmit a single-lead electrocardiogram (ECG), essentially equivalent to lead I of a medical grade 12-lead ECG. This single-lead tracing is recorded by use of two electrodes; one on the back of the watch in contact with the skin and the other on the crown on which the contralateral index finger is placed [3].

These single-lead smartwatch ECGs have been shown to be accurate and provide a significant amount of information, such as heart rate, cardiac conduction, as well as rhythm. All ECG-enabled smartwatches are equipped with software applications that allow the automatic diagnosis of atrial fibrillation [4]. High accuracy of AF diagnosis has been reported based on validation in a limited number of patients with similar clinical profiles. Recent research shows that automatic diagnosis by the smartwatch is challenging in patients with co-existing ECG abnormalities [5,6,7,8]. In addition, more and more consumers present with diagnostic ECG tracings of clinically relevant paroxystic arrhythmias acquired using a smartwatch [1,9]. There is also increasing interest in the potential clinical benefit of interpreting smartwatch ECGs for other cardiac symptoms, such as chest pain, monitoring of medication effects, and detection of anomalies associated with sudden cardiac death [10,11].

Standard use of the Apple Watch solely detects cardiac electrical activity in the direction of lead I. Enlarging the diagnostic spectrum would require the recording of additional ECG leads by sequentially placing the smartwatch on different parts of the body [12]. When comparing the AW tracings with a 12-lead ECG, the standard limb leads (I, II, and III) were found to be equivalent, and the precordial leads (AW-V1 to AW-V6) proved to be very similar [13,14]. Analyzing the electrical activity of the heart from different spatial locations is a cornerstone for the detection of cardiac disorders such as myocardial infarction, pericarditis, Brugada syndrome, or hypertrophic cardiomyopathy but also some forms of arrhythmia such as atrial flutter and pre-excitation [15,16]. A recent study by Sprenge et al. suggested the excellent feasibility and reliability of smartwatches for obtaining precordial lead electrocardiogram recordings. However, only healthy subjects were included, reflecting similar ECG waveform durations and amplitudes. Moreover, the measurements were done by hand on printed ECGs which affects reliability. Lastly, the methods were limited to correlations, unfit for validation purposes [13].

In the current study, we aim to evaluate the reliability of using a smartwatch for obtaining Einthoven leads I, II, III, and precordial leads V1, V3, and V6 in both subjects with and without cardiac abnormalities.

## 2. Materials and Methods

Our institutional review board authorized the research, and the participants gave written informed consent. At rest conditions, participants underwent a standard 12-lead electrocardiogram, followed by up to six smartwatch electrocardiogram recordings using the Apple Watch Series 4 (Apple Inc., Cupertino, CA, USA). For analysis, the first complete tracing (30 s) was used.

### 2.1. Patient Population

A total of 200 individuals with or without a history of cardiovascular disease who presented to the Bordeaux University Hospital’s arrhythmia outpatient and emergency departments were included. Sixty-seven participants were included who had no known cardiac anomalies. In order to introduce a maximal amount of variation in ECG parameter amplitudes and durations, a wide array of cardiac anomalies were included. Fifty-four patients who presented with chest pain were included (23 STEMI, 13 NSTEMI, 5 pericarditis, and 13 of other etiology (e.g., obstructive intracardiac mass, atrial fibrillation, abdominal symptoms)). Furthermore, 56 patients with known cardiac abnormalities associated with sudden cardiac arrest (SCA) were included. This group included ventricular pre-excitation, the Brugada pattern, long QT, and signs suggestive of hypertrophic cardiomyopathy (HCM) and arrhythmogenic right ventricular cardiomyopathy/dysplasia (ARVC/D). To be included, these patients with known clinical diagnoses had to have the following 12-lead ECG characteristic abnormalities: corrected QT interval >480 ms (long QT), Sokolow index >35 mm (HCM), short PR interval + delta wave (ventricular pre-excitation), elevated ST segments type 1 in V1, V2, or V3 (Brugada syndrome) and/or negative T-wave in V1, V2, or V3 ± epsilon (ARVC/D). Lastly, 23 patients with atrial fibrillation, 7 patients with atrial flutter, and 18 patients with bundle branch block were included.

### 2.2. ECG Recordings

The 12-lead ECGs were registered using a 25 mm/s paper running speed and 10 mm/mV amplitude. After short instructions from the healthcare staff, the patient held the AW for 30 s with the back of the AW touching the skin and the right index finger touching the digital crown. Einthoven lead I (AW-I) was recorded with the watch on the left wrist and the right index finger on the crown. Subsequently, Einthoven lead II (AW-II) was recorded with the watch on the left ankle and the right index finger on the crown. Then Einthoven lead III (AW-III) was recorded with the watch on the left ankle and the left index finger on the crown. The locations of V1 (AW-V1), V3 (AW-V3), and V6 (AW-V6) were finally recorded, similar to Wilson’s chest leads, see Figure 1.

It is important to note that precordial smart-watch leads are fundamentally different from 12-lead ECG precordial leads because the precordial electrodes are compared to a single electrode rather than a combination of electrodes. As the best alternative smartwatch, bipolar chest leads are obtained by placing the right index finger on the crown and the watch in the usual precordial location.

### 2.3. Data Processing

All ECGs recorded on an Apple Watch were saved digitally using an iPhone Series 7’s Health app (Apple Inc., Cupertino, CA, USA). A PDF document of each ECG lead was saved for analysis using the “send PDF to your doctor” option and coupled with the de-identified 12-lead ECGs. Two blinded cardiologists diagnosed all patients based on the 12-lead ECGs. A blinded investigator performed the amplitude and duration measurements of the Einthoven and precordial leads of both the 12-lead ECGs and the AW-ECGs used in this study. The amplitude (in millivolts) of the P-waves, QRS-complexes, and T-waves, as well as the duration (in milliseconds) of the PR-intervals, QRS-complex, and QT-intervals were determined (Figure 2).

The amplitudes and durations of the relevant ECG components for the 12-lead ECG were determined from scanned or digital ECGs using digital calipers. For the AW tracings, a custom script was written in Python^®^ (version 3.9.4), which converted the PDF files to digital signals stored as Scalable Vector Graphics (SVG). The signals were then plotted, and calipers were placed on three complexes to determine the mentioned parameters, see Figure 3.

### 2.4. Statistical Analysis

Continuous variables are presented as mean ± standard deviation (SD). Categorical variables are shown as absolute numbers (%). Statistical analyses are performed using Python^®^ (version 3.9.4). Because we are assessing the differences between two measurements of the same substandard, we study the differences, not the agreement. The absolute offset, bias, limits of agreement, and Bland-Altman analyses are used to calculate the differences between the 12-lead and Apple Watch measurements. The bias is defined as the average difference between the standard ECG and the AW. The absolute offset is defined as the average absolute difference. The upper and lower limits of agreement are defined as the estimate of the interval within which a proportion, here 95%, of the differences between the measurements lies. Bland-Altman method was used [17] to compare the amplitudes of the P-waves, QRS-complexes, and T-waves, as well as the PR-intervals, QRS-complex duration, and QT-intervals. A one-sample, two-tailed *t*-test is employed to assess the statistical significance of bias. *p*-values < 0.05 were considered statistically significant.

## 3. Results

### 3.1. Baseline Characteristics

Table 1 summarizes the baseline patient characteristics. The study’s 200 participants (19% female) with a mean (SD) age of 46 (20.0) years underwent simultaneous standard 12-lead and AW recordings. Sixty-seven (33%) of the subjects were without known cardiac, and 133 (67%) had been diagnosed with cardiac disease. Of the 1089 AW tracings, 74 (6.7%) were rejected because of too low signal quality.

### 3.2. Amplitude and Duration of ECG-Waves

A paired *t*-test was conducted to examine differences in mean electrocardiogram (ECG) characteristics between groups. The results indicated that there were no significant differences in the mean amplitudes of the P-wave, T-wave, and ST-segment between groups, with *p*-values of 0.90, 0.48, and 0.65, respectively. However, there was a significant difference in the mean amplitude of the R-wave (*p* < 0.05). In contrast, the mean durations of the PR-interval, QRS-complex, and QT-interval were found to be significantly different between the groups, with *p*-values of <0.001 for all three measures. Amplitudes between 12-lead ECG and the AW matched well with average absolute offsets up to 0.033 mV for the P-wave, 0.33 mV for the QRS-wave, and 0.18 mV for the T-waves. Moreover, for the duration, the measurements showed good agreement between the 12-lead ECG and the AW with average absolute offsets up to 32 ms for the PQ intervals, 26 ms for the QRS interval, and 35 ms for the QT interval. However, the precordial leads did overestimate the R-wave amplitude when compared to the 12-lead ECG, as shown by the significant bias of +0.094 mV for V1, +0.149 mV for V3, and +0.129 for V6 (all *p* < 0.001). Table 2 shows the bias, lower and upper limits of agreement, and absolute offset between the 12-lead ECG and the AW for each ECG amplitude of each lead. Table 3 shows the duration measurements. Figure 4 displays the Bland-Altman plot of the four amplitude parameters (P-wave, R-wave, T-wave, and ST-segment). Figure 5 shows the Bland-Altman plot of the three duration parameters (PR interval, QRS complex, and QT interval). Table 4 shows the linear regression of the Bland-Altman analysis per ECG characteristic by means of the slope and intercept of the line, R-squared (the coefficient of determination R), and the *p*-value.

## 4. Discussion

This validation study aims to investigate the accuracy of emulating V1 through V6 leads using a differential electrode on an Apple Watch for measuring essential parameters in subjects without known cardiac and patients with underlying heart problems. We showed that amplitude and duration measurements are comparable between Apple Watch ECGs and concordant 12-lead ECGs. This provides the first steps toward expanding the range of identifiable cardiac conditions beyond the standard available functionality offered by the current Apple Watch, namely the detection of atrial fibrillation using lead I. Expanding ECG registrations beyond the wrist holds great potential for recognizing symptomatic and asymptomatic disorders and allows for earlier intervention and treatment.

### 4.1. Main Findings

Our results reveal that the smartwatch reliably registers standard Einthoven leads I, II, and III and precordial leads V1, V3, and V6, highly comparable to traditional 12-leads ECGs. Although this is not the first study to show that handheld ECG recorders can record leads other than lead I, it stands out for the following reasons. Firstly, this is, to the best of our knowledge, the first study to quantitatively compare smartwatch ECG with standard 12-leads ECG in a highly diverse group of patients, not just in healthy subjects in sinus rhythm, examples can be seen in Figure 6.

Secondly, previous studies quantitatively comparing smartwatch devices did not include the amplitude of the ST segment [13,18]. This is important as we hope this study will not only encourage the use of wristwatch ECG in the identification of arrhythmias but may also assist in the early detection of myocardial infarction in patients with acute coronary syndrome symptoms [19].

Thirdly, this was the first study to assess quantitative comparisons between different ECG segments from smartwatch recorded Einthoven and precordial leads with those from the standard 12-leads ECG, using a custom solution using digital calipers, based on bias, absolute offset, and 95% limits of agreement as opposed to primary correlation. A previous study, which is most in line with this current study by Sprenger et al., looked solely at correlation coefficient, *p*-values, and bias using an ECG ruler on printed ECGs and AW ECGs [13].

Our study found excellent agreement between the AW and the standard ECG amplitudes and durations. Our Bland-Altman analyses revealed a mostly positive bias. This positive bias suggests that the smartwatch slightly overestimates the amplitude and duration of ECG characteristics. This bias is mainly prominent in the precordial leads when looking at the amplitudes of the ECG characteristics and mainly prominent in the R-wave and T-wave. Showing only significant differences at the R waves in the precordial leads V1, V3, and V6. An example of this overestimation can be seen in Figure 7. The absolute offset for the R-wave and T-wave is more pronounced for the precordial leads. These findings are in line with a recent study on the validity of using the Apple Watch for screening and monitoring QT prolongation, which also found no clinically significant differences between recordings by the Apple Watch and recordings by a standard ECG. The results of this study, however, were limited to the QTc interval and T-wave amplitude in three leads: I, II, and V6 [18]. In two separate studies, the Department of Electrophysiology, Heart Center Leipzig at the University of Leipzig, first studied the reliability of a smartwatch to obtain the Einthoven leads and later the reliability of a smartwatch to obtain the precordial leads, in a similar way this study does [13,14]. These two studies show the same trend of slight overestimation by the AW.

### 4.2. Previous Studies

Several studies have shown proof of principle on the utility of smartwatch ECGs beyond the detection of atrial fibrillation. For example, studies have looked at the use of smartwatch ECGs in pediatric patients, their accuracy in capturing other arrhythmias in adults, and the ability of smartwatches to detect ST-segment changes [20,21]. Previous studies on the validity of amplitude and wave duration of smartwatch tracings only included data from adult participants with sinus rhythm [13,14]. Our current study paves the way for these possibilities and is, to our knowledge, the first study to quantitatively analyze the agreement between a smartwatch and a standard 12-lead ECG in both Einthoven leads (I, II, III) and Wilson leads (V1, V3, V6) in patients with various ECG abnormalities.

The conventional AW-ECG, a single-lead (lead I) measurement, provides a limited view of cardiac electrical activity compared to standard 12-lead ECGs, compromising their ability to detect abnormalities. Although traditional lead I tracings can be useful, they have a high false-negative rate which can provide false reassurance and potentially delay diagnosis and treatment [22]. The diagnostic value of this technique can be increased by using an approach that involves five additional recordings from unconventional positions. For example, research has shown that the inclusion of each of the three precordial tracings increased the diagnostic yield: AW-V1 for Brugada patterns, AW-V1 and AW-V3 for ARVC/D, and AW-V6 for certain cases of HCM, long QT, and ventricular pre-excitation [15].

### 4.3. Study Limitations

There are several limitations to this present study. Firstly, during the recording of the AW-ECG, a physician or researcher was present; therefore, no statement can be made about the quality or location of AW-ECG that would be obtained when a patient makes a recording completely independently. However, the recording of a smartwatch ECG is relatively simple, and other studies show that registrations made by patients outside of the hospital are of high quality [10]. After a brief training session, patients had no difficulty self-recording, according to multiple studies [14]. The steps required to record wrist or precordial ECG tracings are relatively simple, and we suspect that they can be effectively shown in an instructional manual or video, even on the watch [15]. In our study, only tracings obtained by an Apple Watch Series 4 were used, so the findings cannot be generalized and transferred to other smartwatches [23]. The lack of availability of the leads aVR, aVL, and aVF when using the smartwatch could reduce the sensitivity and positive predictive value for the diagnosis of, for example, acute myocardial infarction and its localization. Follow-up research should look at alternative methods of obtaining these missing leads. Further research should also focus on electrode placement to obtain a higher-quality recording. In this study, for the three-lead ECGs, the left lower electrode was placed on the ankle. Research by Pahlm et al. showed the Lund electrode placement system; electrodes are positioned on the major trochanter of the femoral bone and on the lateral arms at the level of the axillary fold to reduce artifacts [24].

### 4.4. Future Clinical Perspective

Solutions are being investigated whereby missing leads can be synthesized by reconstruction of leads. Recent studies also evaluated the possibility of calculating the 12 leads of a standard ECG by recording only 1, 3, or 7 leads, paving the way for a simpler and faster approach to obtaining a full 12-lead ECG recording using an AW [22,23].

Our study provides evidence that tracings obtained with an AW hold great potential as an addition to conventional ECG devices. We propose smartwatch software for ECG tracking should include a feature that clearly indicates the lead being measured. This information should also be included in the PDF output of the ECG to improve the accuracy and interpretation of the data by healthcare professionals and algorithms.

Millions of consumers now have permanent access to a multi-lead ECG machine. The increased accessibility of AW-ECG recordings may facilitate the diagnosis of conditions that may have otherwise gone unnoticed, constituting a leap forward in preventative cardiac medical care. According to research, smartwatches can also record an ECG in newborns and young children with the potential to detect clinically significant conditions. Automating the detection of such findings could provide opportunities to screen and monitor for specific heart diseases as adolescents use smartwatches at an increasing rate [25,26]. This could be extremely beneficial in relatively rare conditions that predispose to SCA, such as Wolff–Parkinson–White syndrome, Brugada syndrome, long-QT syndrome, HCM, and ARVC/D [12].

Smartwatches could also play a major role in the development of telemedicine. Telemedicine is an emerging field that offers an excellent opportunity to improve healthcare with a more individualized approach. One aspect of this development is the availability of technology, which allows for a more flexible approach to healthcare [24]. However, for a smartwatch to become a routine tool for cardiovascular diagnostics and monitoring of disease progression, care must be taken to ensure that the device is as simple and error-free as possible.

Patients may record an ECG independently, anytime, anywhere, due to smartwatch technology. This has the potential to transform our approach to cardiac emergencies and save time and money for both the patient and the healthcare provider. To achieve this, some logistical elements of implementing a smartwatch-based ECG screening program must be explored, including patient confidentiality and health information privacy, data ownership, and a variety of medicolegal implications. Smartwatch-based techniques, like other comparable screening tools, would require careful consideration of how to appropriately integrate them into shared decision-making models, as well as a knowledge of their possible harmful influence on otherwise healthy and asymptomatic persons.

The optimal method for interpreting these tracings is still under development. One option is for the interpretation to be performed by a healthcare professional, which may represent a significant workload, especially in the future, as the popularity of smartwatches with ECG capability increases. Another option is diagnostic classification by artificial intelligence, which is already being used for the detection of atrial fibrillation. An automated screening system, which in selected cases would require review and possibly more specific investigation, seems to be the most feasible at this time.

Even though further research is certainly required, our study has shown that new smartwatch technology in the near future may offer tremendous potential for a quickly available and uncomplicated method to monitor cardiac activity in high-risk patients and diagnose conditions that might have been missed because of the difficulty of getting an ECG.

## 5. Conclusions

Our study shows that an AW is valid in obtaining the Einthoven (lead I, II, and III) and emulating precordial leads (V1, V3, and V6) in a diverse group of subjects with or without cardiac disease. However, further research is needed to establish the clinical utility of the AW, including its accuracy in detecting specific cardiac abnormalities or conditions and its ability to improve patient outcomes.

## Figures and Tables

**Figure 1 sensors-23-02555-f001:**
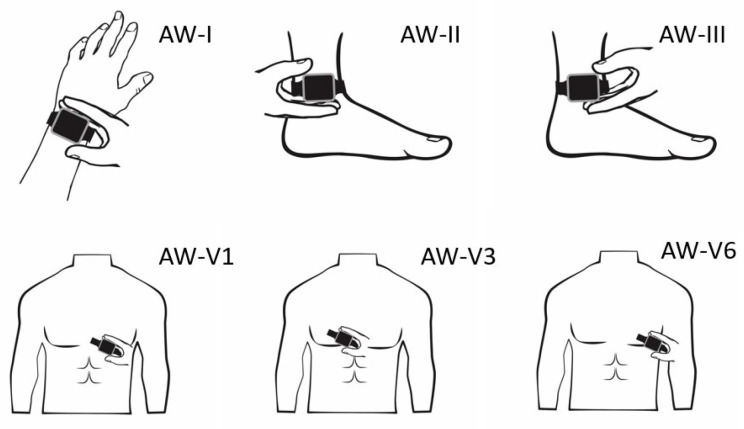
The positioning of the Apple Watch for obtaining the Einthoven and precordial leads.

**Figure 2 sensors-23-02555-f002:**
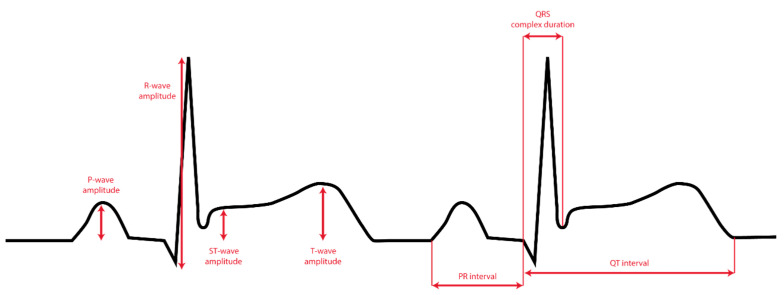
The seven ECG parameters used for the comparison between AW ECG and 12-lead ECG.

**Figure 3 sensors-23-02555-f003:**
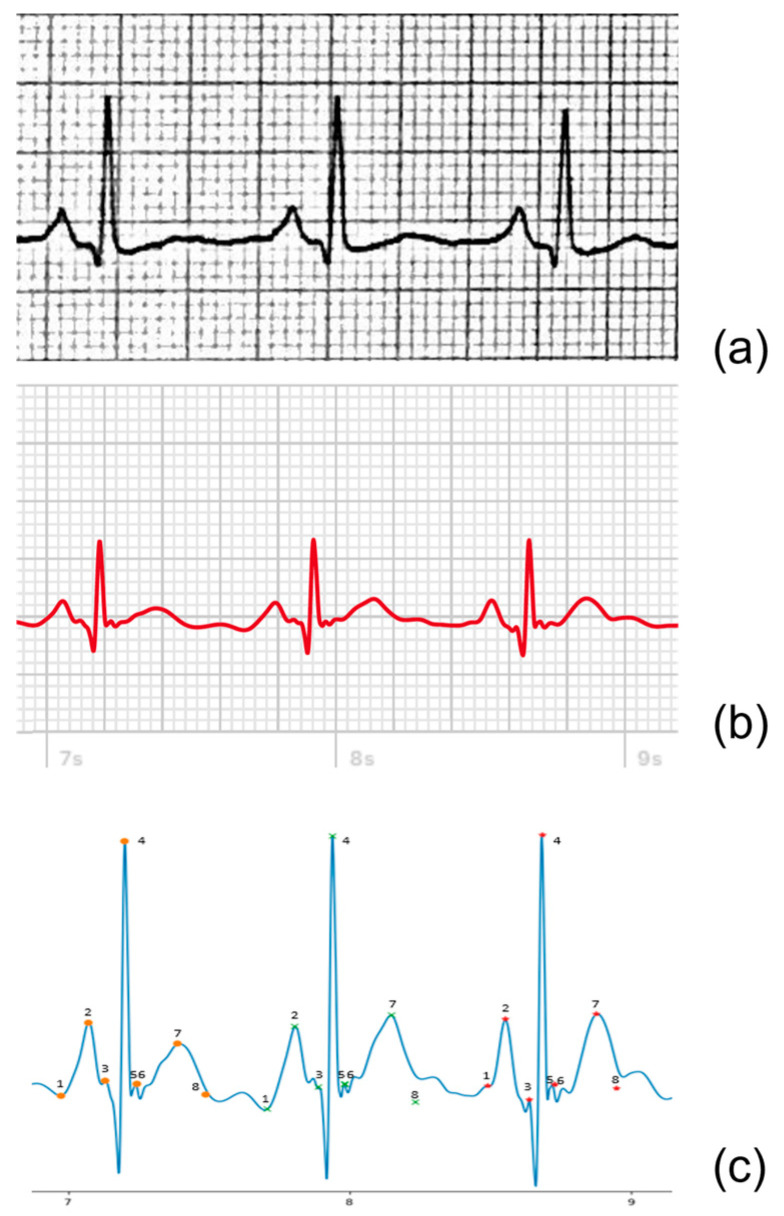
Example of lead II of a 12-lead ECG (**a**), AW-II PDF extract (**b**), and the extracted digital AW-II SVG signal (**c**) derived from the same patient. In the digital tracing (**c**), for each of the three heartbeats, eight points are indicated, from which the four amplitudes and three durations are then automatically determined. The eight points are defined as follows: 1. Onset P-wave, 2. Amplitude P-wave, 3. Onset QRS-complex, 4. Amplitude QRS-complex, 5. End QRS-complex, 6. Amplitude ST-segment, 7. Amplitude T-wave, 8. End T-wave.

**Figure 4 sensors-23-02555-f004:**
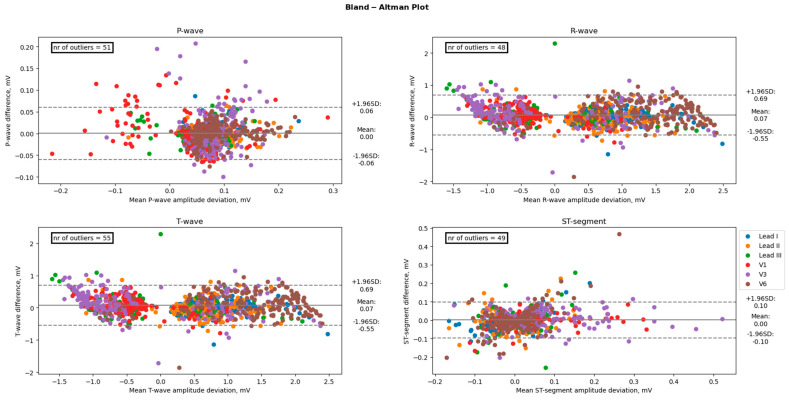
Comparison of the amplitudes between an Apple Watch and the standard 12-lead Electrocardiogram. The black line represents the bias (mean difference), and the dashed lines represent the upper and lower limits of agreement (1.96 SD). The number of points outside these limits is indicated. These differences are considered clinically significant.

**Figure 5 sensors-23-02555-f005:**
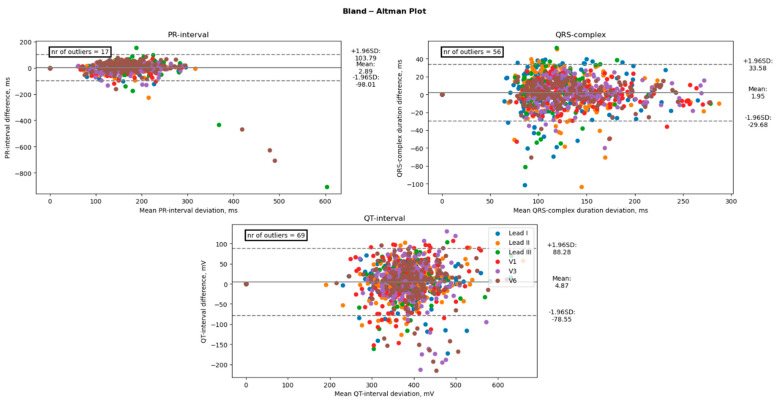
Comparison of the duration between an Apple Watch and the standard 12-lead Electrocardiogram. The black line represents the bias (mean difference), and the dashed lines represent the upper and lower limits of agreement (1.96 SD). The number of points outside these limits is indicated. These differences are considered clinically significant.

**Figure 6 sensors-23-02555-f006:**
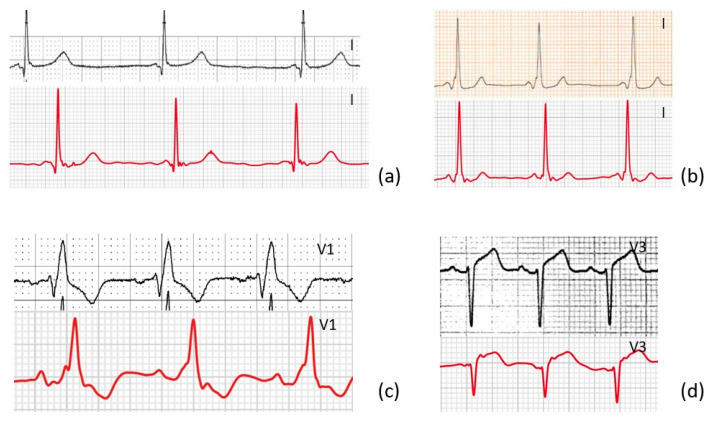
Four examples show a lead from the standard 12-leads ECG and the corresponding AW-lead of ECG-abnormalities from a diverse group of patients. (**a**) Lead I of a patient with long QT syndrome, (**b**) lead I of a patient with WPW, (**c**) lead V1 of a patient with right bundle branch block, (**d**) lead v3 of a patient with ST-elevation.

**Figure 7 sensors-23-02555-f007:**
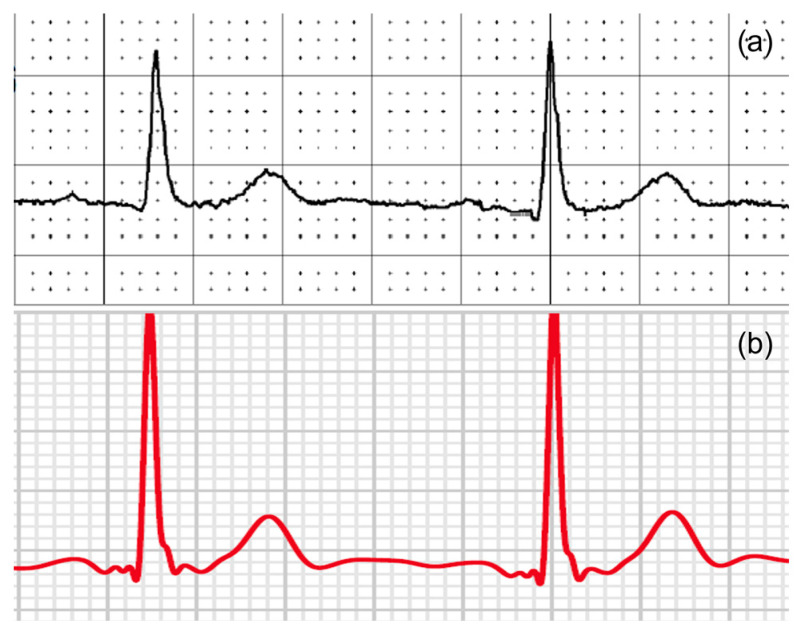
An example of (**a**) V6 from a 12-lead ECG and (**b**) an AW-V6 tracing of the same patient with larger T-waves and R-waves amplitudes on the smartwatch.

**Table 1 sensors-23-02555-t001:** Baseline patient characteristics.

Variables, Units	n
Age (year), mean ± SD	46 (20.0)
Gender (% Female)	19%
No ECG abnormalities	67
STEMI	23
NSTEMI	13
Pericarditis	5
Pre-excitation	12
Brugada	6
Long-QT	4
HCM	4
ARVC/D	5
Atrial fibrillation	23
Atrial flutter	7
Left bundle branch block	10
Right bundle branch block	8
Other patients with chest pain	13

**Table 2 sensors-23-02555-t002:** Electrocardiogram characteristics amplitudes in mV.

Variables, Units	Lead	Mean 12-Leads	Mean AW	Bias	*p*-Value Bias	Lower LoA	CI_Lowerloa	Upper LoA	CI_Upperloa	Absolute Offset
P-wave (mV), mean	I	0.062	0.064	−0.002	0.095	−0.035	[−0.041 −0.030]	0.031	[0.025 0.037]	0.012
	II	0.081	0.083	−0.002	0.090	−0.033	[−0.038 −0.027]	0.028	[0.023 0.034]	0.011
	III	0.048	0.052	−0.004	0.103	−0.050	[−0.058 −0.042]	0.042	[0.034 0.050]	0.018
	V1	0.034	0.034	−0.001	0.851	−0.082	[−0.097 −0.068]	0.081	[0.067 0.095]	0.032
	V3	0.084	0.079	0.005	0.130	−0.087	[−0.103 −0.071]	0.098	[0.081 0.114]	0.033
	V6	0.095	0.092	0.002	0.122	−0.039	[−0.046 −0.032]	0.044	[0.036 0.051]	0.016
R-wave (mV), mean	I	0.763	0.755	0.009	0.534	−0.364	[−0.429 −0.298]	0.381	[0.316 0.446]	0.138
	II	0.481	0.494	−0.013	0.528	−0.560	[−0.656 −0.464]	0.533	[0.437 0.629]	0.208
	III	−0.187	−0.236	0.049	0.209	−0.683	[−0.811 −0.555]	0.781	[0.652 0.909]	0.232
	V1	−0.400	−0.494	0.094	1.47 × 10^−6^	−0.382	[−0.466 −0.299]	0.571	[0.487 0.654]	0.201
	V3	−0.158	−0.307	0.149	8.61 × 10^−7^	−0.630	[−0.766 −0.493]	0.928	[0.792 1.065]	0.332
	V6	1.347	1.218	0.129	1.19 × 10^−6^	−0.562	[−0.683 −0.441]	0.820	[0.698 0.941]	0.296
T-wave (mV), mean	I	0.124	0.129	−0.005	0.479	−0.196	[−0.230 −0.163]	0.186	[0.153 0.220]	0.069
	II	0.162	0.169	−0.007	0.085	−0.107	[−0.125 −0.090]	0.094	[0.076 0.112]	0.038
	III	0.061	0.079	−0.018	0.081	−0.210	[−0.243 −0.176]	0.174	[0.141 0.207]	0.074
	V1	0.117	0.107	0.010	0.265	−0.219	[−0.259 −0.179]	0.239	[0.200 0.280]	0.087
	V3	0.384	0.397	−0.013	0.505	−0.523	[−0.612 −0.433]	0.497	[0.408 0.587]	0.182
	V6	0.295	0.285	0.010	0.131	−0.175	[−0.208 −0.143]	0.196	[0.164 0.230]	0.069
ST- segment (mV), mean	I	−0.009	−0.004	−0.005	0.065	−0.077	[−0.090 −0.065]	0.067	[0.055 0.080]	0.026
	II	−0.001	0.000	−0.001	0.781	−0.092	[−0.107 −0.076]	0.090	[0.074 0.105]	0.031
	III	0.008	0.008	0.000	0.967	−0.120	[−0.141 −0.099]	0.120	[0.099 0.141]	0.039
	V1	0.040	0.035	0.005	0.088	−0.065	[−0.077 −0.052]	0.074	[0.062 0.086]	0.026
	V3	0.057	0.051	0.006	0.087	−0.088	[−0.104 −0.071]	0.100	[0.083 0.116]	0.037
	V6	0.000	0.000	0.000	0.959	−0.127	[−0.149 −0.104]	0.127	[0.105 0.150]	0.039

**Table 3 sensors-23-02555-t003:** Electrocardiogram characteristics durations in milliseconds.

Variables, Units	Lead	Mean 12-Leads	Mean AW	Bias	*p*-Value Bias	Lower LoA	CI_Lowerloa	Upper LoA	CI_Upperloa	Absolute Offset
PR-interval (msec), mean	I	153.6	149.8	3.8	0.1	−52.0	[−61.7 −42.2]	59.6	[49.8 69.4]	21.0
	II	148.4	144.5	3.9	0.1	−54.3	[−64.5 −44.1]	62.1	[51.9 72.3]	18.8
	III	79.1	78.1	0.9	0.9	−151.3	[−178.0 −124.6]	153.2	[126.5 179.9]	22.5
	V1	139.9	136.6	3.3	0.1	−49.9	[−59.2 −40.6]	56.5	[47.1 65.8]	18.2
	V3	157.8	155.5	2.3	0.3	−55.0	[−65.1 −45.0]	59.6	[49.5 69.6]	20.4
	V6	162.5	159.3	3.2	0.6	−155.8	[−183.7 −127.9]	162.1	[134.3 190.0]	32.5
QRS-complex (msec), mean	I	116.8	113.9	2.8	0.1	−39.5	[−47.0 −32.1]	45.2	[37.8 52.6]	16.9
	II	111.1	108.5	2.6	0.1	−36.4	[−43.3 −29.6]	41.7	[34.8 48.6]	14.8
	III	55.8	53.8	2.0	0.1	−26.8	[−31.8 −21.7]	30.7	[25.7 35.8]	7.4
	V1	122.4	121.0	1.4	0.1	−24.6	[−29.1 −20.0]	27.4	[22.9 32.0]	9.5
	V3	134.0	132.5	1.5	0.1	−23.8	[−28.2 −19.4]	26.8	[22.4 31.3]	9.5
	V6	124.9	123.6	1.3	0.1	−21.8	[−25.8 −17.7]	24.4	[20.3 28.4]	8.5
QT-interval (msec), mean	I	371.1	367.1	4.0	0.1	−67.1	[−79.5 −54.6]	75.1	[62.7 87.6]	25.9
	II	334.3	330.0	4.3	0.1	−67.7	[−80.3 −55.1]	76.4	[63.8 89.0]	26.5
	III	183.0	178.8	4.2	0.1	−60.0	[−71.2 −48.7]	68.3	[57.1 79.6]	18.1
	V1	324.0	318.5	5.5	0.1	−86.6	[−102.7 −70.4]	97.6	[81.5 113.8]	35.4
	V3	367.5	362.3	5.2	0.1	−93.9	[−111.3 −76.5]	104.2	[86.9 121.7]	34.9
	V6	377.0	371.1	5.9	0.1	−89.3	[−106.0 −72.6]	101.2	[84.5 117.8]	33.3

**Table 4 sensors-23-02555-t004:** Linear regression characteristics Bland-Altman analysis.

	Slope	Intercept	R-Squared	*p*-Value
P-wave	0.84	0.01	0.67	<0.05
R-wave	0.99	−0.07	0.90	<0.05
T-wave	0.89	0.02	0.76	<0.05
ST-segment	0.67	0.00	0.60	<0.05
PR-interval	0.90	11.57	0.64	<0.05
QRS-complex	0.96	2.18	0.93	<0.05
QT-interval	0.95	12.51	0.92	<0.05

## Data Availability

The datasets generated during during the current study are available from the corresponding author on reasonable request.

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
