# Peer review of "Using a Smartwatch to Record Precordial Electrocardiograms: A Validation Study"

_sensors, 2023, doi:10.3390/s23052555_

Round 1
Reviewer 1 Report
The authors present a study of mimicking the standard ECG leads using Apple Watch in some non-standard modes of operation. They confirm the applicability of the instrument for use in clinical applications.
I have a few remarks regarding the manuscript:
1. In Abstract you start with: "Smartwatches which support recording of a single-lead electrocardiogram (ECG) are increasingly being used beyond the wrist."
I really did not understand what "beyond the wrist" was alluding to until I read the article, which was a bit annoying, so I would implore the authors to define exactly what they mean already in the abstract.
2. You acknowledge that "precordial smart-watch leads are fundamentally different", but then act as if they were identical to standard precordial leads in the analysis. E.g. in Figures 6 and 7 you show the significant differences between the two. "Fundamentally different" is really the key here, as one should not expect a close similarity between the emulated and real precordial leads. I believe it should be further stressed that you are comparing apples to oranges in the later analysis too. The question posed here is not even whether Apple watch can measure V1 through V6 but rather how well can these leads be emulated using a differential electrode. It still makes total sense to test the presented cases but perhaps it could be stressed that the differences are expected.
3. The conclusions are too strong. You write that you "demonstrated the validity of an AW to adequately obtain the [leads]" - adequately for what? Even if statistics of singular measures seem ok (which they do not for precordial leads), one cannot claim that these results are sufficient for broad clinical application. Please focus your conclusions better.
Reviewer 2 Report
This manuscript presents a new study on using Apple watches to register electrocardiograms (ECG), other than the single-lead electrocardiogram. The idea is to determine a procedure to capture patient status using these devices and verify their reliability in measuring frontal and precordial ECGs other than lead I. As the authors said, the clinical validation study assessed the reliability of an Apple Watch (AW) to obtain conventional frontal and precordial leads compared to standard 12-lead ECGs in subjects without known cardiac anomalies and patients with underlying heart disease. They studied 200 subjects (67% with ECG anomalies). The authors could validate the reliability of such devices to obtain useful results that may indicate their future incorporation into routine medical care.
The authors did a good job. The experiment is well described and will certainly advance the state-of-the-art in this specific application of sensors in healthcare. Despite that, I have some suggestions that may improve the final version of this manuscript.
· I missed a table synthesizing the main desired characteristics of the SW ECG observed in the related publications.
· An in-depth evaluation of the main proposals would enrich the present manuscript.
· Another table comparing the obtained results with others published would help users understand how far the authors' proposal is.
The suggestions are not mandatory. I enjoyed reading this manuscript and believe it will contribute to many other groups researching the same topic.
Reviewer 3 Report
I revised with interest Joske vander Zande's manuscript, since more and more tools are being utilized by individuals to diagnose rhythm disorders. Bland-Altman method permits the measurement of bias between the two techniques, hence the authors' choice of the Apple Watch validation method was a wise one.
However, in order for the results to be appropriate for publishing, the authors must supply further information:
1. Please supply the mean values of the variables derived by the two methods: P wave, R wave, and T wave amplitude in mV, PR-interval, QRS complex, and QT interval duration in milliseconds.
2. Are the means calculated at point 1 for the two ECG evaluation methods statistically different?
3. Please generate a linear regression using differences and means for the two ECG methods.
4. Figure 5. Please mention the number of values beyond the 1.96 SD lines (at least 40 values are marked below -1.96 SD line). The figure depicts very wide limits of agreement that are likely to be highly clinically relevant.
